

# Topographic changes due to the 2004 Chuetsu thrusting earthquake in low mountain region

Zhikun Ren[1,2], Takashi Oguchi[3], Peizhen Zhang[2], Shoichiro Uchiyama[4]

*1 Key Laboratory of Active Tectonics and Volcanos, Institute of Geology, China Earthquake*

*Administration, P.O. Box 9803, Beijing 100029, China; rzk@ies.ac.cn*

*[2]State Key Laboratory of Earthquake Dynamics, Institute of Geology, China Earthquake*

*Administration, P.O. Box 9803, Beijing 100029, China*

*[3]Center for Space Information Science, The University of Tokyo, Chiba, 2778568 Japan*

*[4]National Research Institute of Earth Science and Disaster Prevention, Tsukuba, Ibaraki,3050006*

*Japan*

*Corresponding address:

    State Key Laboratory of Earthquake Dynamics

    Institute of Geology

    China Earthquake Administration

    Beijing 100029, China

    Email:**rzk@ies.ac.cn**

        **lzkren@gmail.com**



**Abstract**

The co-seismic landslide volume information is critical to understanding the role of strong earthquake

in topographic evolution. However, the co-seismic landslide volumes are mainly obtained using

statistical scaling laws, which are not accurate enough for quantitative studies of the spatial pattern of

co-seismically induced erosion and the topographic changes caused by the earthquakes. The availability

of both pre- and post- earthquake high-resolution DEMs provide us the opportunity to try new approach

to get robust landslide volume information. Here, we propose a new method in landslide volume

estimate and tested it in Chuetsu region, where a Mw 6.6 earthquake occurred in 2004. Firstly, we align

the DEMs by reconstructing the horizontal difference, then we quantitatively obtained the landslide

volume in the epicentral area by differencing the pre- and post-earthquake DEMs. We convert the

landslide volume into the distribution of average catchment-scale seismically induced denudation. Our

results indicate the preserved topography is not only due to the uplifting caused by fault-related folding

on the hangwall of Muikamachi fault, but also undergone erosion caused by the seismically induced

landslides. Our findings reveal that Chuetsu earthquake mainly roughens the topography in the Chuetsu

region of low elevation. This study also reveal that the differential DEM method is a valuable approach

in analyzing landslide volume, as well as quantitative geomorphic analysis.

Keywords: Chuetsu earthquake; topographic change; LiDAR; Differential DEM; denudation

**1. Introduction**





It is increasingly recognized that the role of tectonic events is critical to understanding topographic
evolution, such as strong earthquakes. Strike-slip earthquakes mainly cause horizontal deformation,
normal fault earthquakes mainly occurred in extensional environment than reduce the topography, and
thrust earthquake is the main one causing surface uplift, hence mountain building. It has been realized
that strong thrust earthquakes play important role in the topographic evolution in regions of steep relief
and high elevation, such as the marginal zones of high plateau at Himalaya (Avouac, 2003; Larsen and
Montgomery, 2012; Morell et al., 2015; Owen, 2010), Longmen Shan (Hovius et al., 2011; Li et al.,
2014; Parker et al., 2011; Ren et al., 2014a) and Andes (McPhillips et al., 2014). Previous studies
demonstrated that the landslides are thought to limit the slope (Blöthe et al., 2015; Burbank et al., 1996)
and height of mountain peaks above adjacent river valleys in steep orogenic regions (Larsen and
Montgomery, 2012; McPhillips et al., 2014; Roering, 2012). Recent studies found that the erosion
caused by landslides did not change much in response to climatic changes; hence, the tectonic events
such as earthquakes are the primary landslide trigger in the arid foothills of Peru in steep Andes
(McPhillips et al., 2014). Quantifying erosion rate is critical to understanding the role of tectonic events
in mountain building. However, due to the long-term mountain building and topographic evolution,
previous studies are mainly regional studies based on sparse thermochronological dating (Kirby et al.,
2002; Wang et al., 2012), cosmogenic dating (Ansberque et al., 2015; Godard et al., 2010; Ouimet,
2010) or modern hydrological observations (Dadson et al., 2003) in region of high mountain area. These





regional studies could not show the details of how tectonic events act in topographic evolution of low
mountain region. Strong earthquakes are the most recent tectonic events, which provide us the valuable
opportunity to study the role of such events in current topographic evolution of low mountain region.
However, the co-seismic landslide volumes are usually obtained using statistical scaling laws, which
has large uncertainties in different regions by applying same scaling laws. Different researchers could
get totally different co-seismic landslide volumes for one earthquake using different methods (Li et al.,
2014; Marc et al., 2015; Parker et al., 2011; Ren et al., 2014b; Ren et al., 2017).
Recently, the high-resolution and multi-temporal Light Detection and Ranging (LiDAR) Digital
Elevation Models (DEMs) or DEM generated from stereo pair of remote sensing images have been
proven valuable in monitoring geomorphic, co-seismic and volcanic surficial deformations (Cowgill et
al., 2012; Lane et al., 2001; Ren et al., 2014a; Stumpf et al., 2014; Wheaton et al., 2010; Zhou et al.,
2015; Zielke et al., 2010). By differential pre- and post-earthquake DEM, we could quantitatively
analysis the topographic changes and evaluate the landslide volume. It has been used to derive
co-seismic landslide volumes in Longmen Shan region by differencing pre- and post- Wenchuan
earthquake DEMs, as well as topographic analysis (Ren et al., 2014a). However, in region of low
mountain, the role of strong earthquakes in topographic evolution is rarely reported. The 2004 Mw 6.6
Chuetsu earthquake occurred in Niigata prefecture in Japan, where the local relief of the epicentral area
is low with maximum elevation of 765 m (Fig. 1). In this study, we use the high-resolution pre- and



post-earthquake DEM (GSI, 2007) to study the topographic changes due to the Chuetsu earthquake, by
comparing the slope angle, slope aspect, relief and roughness pre- and post-earthquake. The co-seismic
denudation distribution pattern was also analyzed using the co-seismic landslide volume with the
availability of multi-temporal high-resolution topographic data.  We finally discussed the role of
earthquake in topographic evolution at Chuetsu area of low mountain.
**2. Tectonic Setting**
The 2004 Mw 6.6 Chuetsu earthquake occurred at Chuetsu, Niigata prefecture Japan, where the
convergent plate boundary between the Amurian and Okhotsk plates is located (Fig. 1, (Okamura et al.,
2007; Okamura et al., 1995; Wei and Seno, 1998)). The epicentral area is of low elevation with
maximum of 765 m, which is composed of sedimentary and volcanic rocks from Holocene to Miocene.
The sediments were mainly formed in the early Miocene, concurrently with the opening of the Japan
Sea (Fig. 1). The sediments have been folded under E-W to WNW-ESE compressional stress field since
~2-3 Ma (e.g., (Hirata et al., 2005; Okamura, 2003)). The continued compression deformed the strata,
landforms and caused the repeated seismicities in Chuetsu area. The Shinano River is the main river
flow through the Chuetsu area where the flood plain is mainly composed of Holocene to Late
Pleistocene sediments (Fig. 2). The mountain area is mainly composed of Pleistocene to Pliocene
sediments, accompanied with a tectonic window composed of Jurassic sediments (Fig. 2). The uplifting
of the mountain is proposed to be due to fault-related folding caused by the thrust along the NS trending



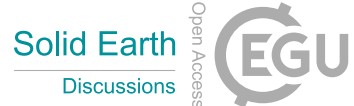

Muikamachi-Bonchi-Seien fault (Fig. 2, )(Kato et al., 2005; Kato et al., 2006; Okamura et al., 2007).
The 2004 Chuetsu earthquake is a thrust-dominated earthquake with minor lateral motion (Maruyama et
al., 2005). It has been reported that there is a co-seismic surface rupture zone of 1 km in length, with
~20 cm vertical co-seismic offset and lateral offset less than 20 cm on a previously unmapped fault
(Maruyama et al., 2005), which lies along the northward extending of the Muikamachi fault (Fig.2 ,
(Nakata and Imaizumi, 2002; RGAFJ, 1991)). Hence, the most possible causative fault of the Chuetsu
earthquake is the Muikamachi fault, according to the focal mechanism and location of surface ruptures
(Maruyama et al., 2005). Previous studies mapped the subsurface fault with detachment in depth of
~10-13 km, which agreed well with the distribution of aftershocks (Kato et al., 2005; Kato et al., 2006;
Okamura et al., 2007). They found that the fault-related folding on the hanging wall of the Muikamachi
fault was responsible for the growth of the geological structures (Kato et al., 2005; Kato et al., 2006;
Okamura et al., 2007; Suppe, 1983). Paleoseismology studies reveal at least two strong earthquakes
occurred in the past 9000 years prior to the occurrence of the 2004 Chuetsu earthquake(Maruyama et al.,
2005). The co-seismic displacements of the two paleoearthquakes were almost identical at ~1.5 m,
which was almost 15 times of the 2004 event (~10 cm). The 2004 Chuetsu earthquake triggered
thousands of co-seismic landslides, which dramatically modified the local topography (Chigira and
Yagi, 2006; Dou et al., 2015; Sato et al., 2005; Wang et al., 2007). Hence, the mountain growth in the
epicentral area should be closely related to the co-seismic landslides caused by repeated strong



earthquakes.

**3. Data and Methods**
**3.1. Data**
The pre-earthquake DEM is of 10 m in resolution with absolute vertical precision within 2.5 meter. The
10-m-resolution DEM is generated from stereo pairs of aerial photographs or topographic maps that
covering the whole Japan area at Geospatial Information Authority (GSI) of Japan (Freely available at
http://fgd.gsi.go.jp/download). The post-earthquake DEM is of 2 m resolution with root-mean-square
(RMS) error within 0.12 m that generated from airborne LiDAR data surveyed in 2005 with point
density larger than 1 pt/m$^2$, released by the GSI of Japan ((GSI, 2007). These DEMs are of higher
precision than that used in our previous studies in Wenchuan area (Ren et al., 2014b) (Fig. 3). The
landslide inventory map is interpreted based on high-resolution aerial photograph, by the National
Research Institute for Earth Science and Disaster Prevention (NIED), Japan (Fig. 4, (Chigira and Yagi,
2006; Dou et al., 2015; Sato et al., 2005; Wang et al., 2007)). The geological information is derived
from the 1:200,000 geological maps provided by the Geological Survey of Japan (GSJ) (Figs. 2 and 4a,
Freely available at https://gbank.gsj.jp/seamless/index_en.html) and the active fault map of Japan (Fig.
1; (Nakata and Imaizumi, 2002; RGAFJ, 1991)).

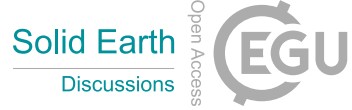

## 131 3.2. Methods

## 132 3.2.1. Differential DEM

With the availability of high-resolution DEM data pre- and post-earthquake, especially the LiDAR
DEM, the differential DEM method are widely used in detecting topographic changes (Chen et al., 2006;
Ren et al., 2014a; Stumpf et al., 2014), co-seismic deformations (Cowgill et al., 2012; Nissen et al.,
2014; Zhou et al., 2015) as well as sediment budgets (Lane et al., 2001; Wheaton et al., 2010) . Previous
studies have shown that the differential DEM method using multiple-scale and multiple-source DEMs is
effective in detecting topographic changes caused by co-seismic landslides (Chen et al., 2006; Ren et al.,
2014a; Ren et al., 2017). The available of the pre- and post-earthquake DEMs in Chuetsu area provide
us the opportunity to study the topographic changes caused by the co-seismic landslides due to the 2004
Chuetsu earthquake. In differential DEM method, the precise georeference and correlation between the
multi-temporal DEMs is one of the key issues before subtracting. To analysis the topographic changes
under compression environment, we are mainly interest of the vertical deformations. The horizontal
differences between the pre-and post-earthquake DEMs were calculated and then reconstructed by
back-slipping the horizontal differences. The cosi-corr software is developed to measure sub-pixel
ground deformation using optical satellite and aerial images, which has an accuracy of 1/10 of the input
pixel size (Ayoub et al., 2015; Hollingsworth et al., 2012; Leprince et al., 2007; Zhou et al., 2015). In
this study, we could estimate the horizontal differences between the pre- and post-earthquake DEMs



using the cosi-corr software (freely available at
www.tectonics.caltech.edu/slip_history/spot_coseis/index.html), following Zhou et al.'s method (Zhou
et al., 2015). The airborne LiDAR DEM was downsampled to 10 m to match the pre-earthquake DEM.
We used a correlation window of 64 pixels followed by 32 pixels with a step of 4 pixels (40 m). The
sub-pixel matching procedure was performed on the frequency content, which is more accurate than the
statistical correlator (Ayoub et al., 2015; Zhou et al., 2015). Consequently, we got the NS (Fig. 3a) and
EW (Fig. 3b) components of the horizontal differences between the pre- and post-earthquake DEMs
(Fig. 3). Then by reconstructing the mean horizontal differences in both directions to the whole DEM,
we obtained the precisely geo-referenced and correlated DEMs. Finally, by differencing the pre- and
post-earthquake DEMs, we obtained the vertical deformations caused by the Chuetsu earthquake (Figs.
3c and 4). The largest landslide clearly shows the source and deposit areas, which occurred in the low
mountain composed of late Miocene to Pliocene non-marine sediments (Fig. 4). Meanwhile, the derived
landslide volumes also show consistent results that deep-seated landslides are the main contributor to
the landslide volumes (Fig. 5).

**3.2.2. Topographic Analyses**
Steep slopes are prone to landslides (Burbank et al., 1996; Dai and Lee, 2002; Densmore et al., 1998),
such as the co-seismic landslides triggered by the 2008 Wenchuan Mw 7.9 earthquake which mainly



occurred on slopes with angles larger than 30° (Ren and Lin, 2010). Previous studies have found that
slope angle, slope aspect, relief and roughness are the four main topographic features widely used in
geomorphological studies, which could be used to analysis the topographic changes due to the
co-seismic landsliding. Statistical comparison of the pre- and post-earthquake topographic features has
been proven to be useful in analyzing the co-seismic topographic changes (Ren et al., 2014a). In this
study, based on the downsampled 10 m resolution pre- and post-earthquake DEMs, we compare the pre-
and post-earthquake slope angle, slope aspect, relief and roughness, respectively (Fig. 6). The
co-seismic displacement is less than 20 cm in the epicentral area, hence, the topographic changes are
mainly due to co-seismic landsliding. Thus, in this study, we statistically compare the topographic
changes within each landslide polygon (Fig. 6).

**3.2.3. Catchment-scale Denudation depth**
Landsliding is the dominant mass wasting process in humid uplands (Hovius et al., 1997), thus, it is
reasonable to derive the denudation using the landslide volumes. The topographic changes within the
landslide area should be much larger than the ~10 cm co-seismic displacements. Thus it is reliable to
derive landslide volume using the subtracted DEM. We calculated the co-seismic landslide volumes in
the epicentral area, by summing the elevation changes within the landslide area and multiplying the
summed value by the area of one pixel (100 m$^2$). There are positive values and negative values for each





185 single landslide, because there are source and deposit areas, correspondingly (Fig. 4b and 4c). Hence we

186 firstly sum both the positive and negative value for each landslide, then summed the absolute value

187 together and finally take the average value as the landslide volume for this landslides. The catchments

188 are derived using the flow accumulation data of the streams and outlet data of each catchment from the

189 pre-earthquake DEM with stream length as short as 5 km (Fig. 7). To obtain distribution pattern of the

190 catchment-scale denudation depth, we summed the landslide volumes within each catchment. The

191 average denudation depth was obtained by dividing the summed landslide volume by the catchment

192 area (Figs. 7-8).

193

194 **4. Results**

195 Using cosi-corr software, we obtained the NS component difference of -0.27 m with standard deviation

196 (STD) of 3.12 m and the EW component difference of 0.21 m with STD of 3.37 m (Figs. 3b and c).

197 Before differential the DEMs, we first reconstruct the corresponding differences of NS and EW

198 components. Then we obtain the elevation changes by subtracting the pre-earthquake DEM from the

199 post-earthquake DEM. The flat ground surfaces that located far from the epicentral area should be

200 stable during the earthquake, i.e., the real elevation changes should be nearly zero. Hence, the obtained

201 elevation differences at such regions represent the accuracy of the differential DEM in this study. The

202 elevation differences ranges from -0.46 m to 0.32 m at the non-deformed flat region shown in figure 3c

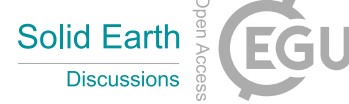

(Fig. 3c). We then obtain the mean elevation change of the whole region of -0.24 m with STD of 2.22 m
(Figs. 3d) by eliminating the elevation changes between -0.46 m and 0.32 m, i.e., setting the delta-z
values within this range to 0. The obtained mean vertical deformation is comparable with the maximum
co-seismic displacement from Interferometric Synthetic Aperture Radar (InSAR) results (Ozawa et al.,
2005) and field investigations (Maruyama et al., 2005). In the landslide region, the elevation changes
could reach tens of meter, which are much greater than 0.32 m and less than -0.46 m, hence it is reliable
to analyze the topographic changes using the pre- and post- earthquake DEMs. The mean elevation
change in the landslide region is 0.08 with STD of 2.17 m (Fig. 4b). As shown in Figure 4c, the
landslide scarp and toe of the largest landslide is clearly shown on the differential DEM map. The
results show the total volume of the 330 deep-seated landslides is ~0.26 km$^3$ (Fig. 5), which is
comparable with the total volume of ~0.30 km$^3$ of the ~6000 shallow landslides (Fig. 5). The
catchment-scale average denudation depth distribution shows maximum denudation of 894 mm, which
did not locate right above the surface ruptures (Figs. 7-8, (Maruyama et al., 2005)). The maximum
denudation correlates with the uplifting pattern in the epicentral area suggested by the fault-related
folding on the hangwall of the Muikamachi fault (Fig. 8, (Kato et al., 2005; Kato et al., 2006; Okamura
et al., 2007)).

The co-seismic topographic changes in the epicentral area show consistent increase in slope angle, relief





and roughness (Fig .6). The comparison of the pre- and post-earthquake topographic features within
each catchment also indicate the average hillslope, relief and roughness are all increased after the
Chuetsu earthquake (Fig .6). The slope aspect decreases in 0°-135° and 270°-360° and increases in
135°-270° (Fig. 6). The observed slope aspect changes might be related with the co-seismic lateral
displacement of the 2004 Chuetsu earthquake, which was reported to be ~10-20 cm (Maruyama et al.,

226   2005).


**5. Discussion**
The occurrence of the Chuetsu earthquake provides a valuable opportunity to quantitatively analysis the
co-seismic topographic changes and denudation caused by co-seismic landslides with the availability of
the pre- and post- earthquake DEM in the epicentral region, hence discussing the tectonic process and
topographic growth in region of low mountain. This study might be the first time to study the
topographic changes using differential DEM method in region of low mountain. The key question in
differential DEM study is the pre- and post- earthquake DEMs are of different sources, which might
represent different surfaces, such as the topographic or bare-earth surface. Usually, the DEM from
stereo pair of images should represent the topographic surfaces including the canopy of the forest, and
the LiDAR derived DEM represents the bare-earth DEM. However, in this study, there are no
systematical elevation errors observed. Meanwhile, the blank area in Fig 4b show the elevation



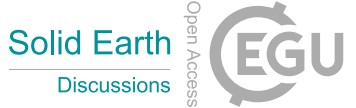

differences beyond -0.46 m to 0.32 m. The canopy of forest could not be such small. Hence, the
differential DEM results should represent the canopy of the forest, which indicate that the pre- and
post-earthquake DEMs are referring the same surface, i.e., our results represent the real co-seismic
elevation differences, however, due to the precision and resolution of the DEMs, it could not show the
co-seismic vertical deformations less than ~20 cm clearly. In this study, based on the pre- and
post-earthquake DEMs, we statistically analyzed the topographic changes caused by the Chuetsu
earthquake in terms of the slope angle (Fig. 7a), slope aspect (Fig. 7b), relief (Fig. 7c), roughness (Fig.
7d) and the catchment-scale denudation (Figs. 7-8). The slope angle, relief and roughness are all
coseismically increased after the Chuetsu earthquake in landslide-scale (Fig. 6) and catchment-scale
(Figs. 8a-c) at the epicentral area (Fig. 6). The slope aspect changes show decrease in 0°-135° and
270°-360° and increase in 135°-270°, which might be associated with the co-seismic lateral
displacement along the NS trending rupture (Maruyama et al., 2005). The comparisons of pre- and
post-earthquake data suggest the Chuetsu earthquake is mainly roughening the topographic relief (Fig.
6), which is consistent with the role of long-term seismic landsliding (Blöthe et al., 2015; Larsen and
Montgomery, 2012; McPhillips et al., 2014; Roering, 2012).
In this paper, the co-seismic landslide volumes are obtained using differential DEM method, then we
convert the landslide volume information into seismically induced erosion. We find that, in the Chuetsu
area, the catchment-scale denudation depth distribution did not show direct correlation with the distance

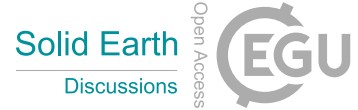

to the surface rupture (Fig. 8). The denudation distribution pattern shows correlation with the uplifting
pattern suggested by fault-related folding on the hangwall of the Muikamachi Fault (Fig. 8, (Okamura et
al., 2007)). The fault-related folding suggest the main uplifting area lies ~8-10 km away from the
Mukamachi fault on the hangwall (Kato et al., 2005; Kato et al., 2006; Okamura et al., 2007; Suppe,
1983). However, in the steep Wenchuan area, previous studies have found that long-term high
denudations are concentrated in a narrow zone along the Longmen Shan Thrust Belt, revealed by
erosion rates from kyr-scale cosmogenic $^{10}$Be and Myr-scale low temperature thermochronology dating
methods (Godard et al., 2010; Kirby et al., 2002; Ouimet, 2010). The highest co-seismic denudation
also mainly concentrated in the narrow corridor between the co-seismic surface ruptures produced by
the 2008 Wenchuan earthquake (Ren et al., 2014a). Hence there might be two possibilities why the
topographic relief profile did not directly correlate with the uplifting pattern in Chuetsu region (Fig. 8).
The first possible reason might be due to the erosion differences. The high erosion occurred in the high
uplifting area due to fault-related folding; but the area near the fault is of low erosion and uplift (Fig. 8).
Thus, the almost homogeneous topographic relief is preserved in the epicentral area under the coupling
process of co-seismic uplift and denudation. The correlation between the denudation and uplifting also
indicates the topographic growth in orogenic belt is closely related with the deformations associated
with the major faults, especially in regions that the uplifting is dominated by fault-related folding
(Okamura et al., 2007; Suppe, 1983). Hence, the strong earthquakes should play important role in the





mountain growth in the epicentral area since initiation of compression and folding at ~2-3 Ma (Hirata et
al., 2005; Okamura, 2003). The second possible reason is that the topography near the Muikamachi fault
might be due to the strong paleoearthquakes associated with large co-seismic vertical offsets as that
revealed by trenching (Maruyama et al., 2007). Paleoseismological studies reveal at least two strong
earthquakes occurred during the past ~9000 years, which produced co-seismic vertical displacement
that are almost 15 times of that produced by the Chuetsu earthquake (Maruyama et al., 2005; Maruyama
et al., 2007). A vertical slip rate of 0.4 mm/yr was calculated using the dating results of the displaced
units in the trench (Maruyama et al., 2007). The Chuetsu area is composed of young sediments mainly
formed since Miocene, which is under continued thrusting and folding since ~2-3 Ma (Hirata et al.,
2005; Okamura, 2003). The total uplifting could be roughly estimated to be ~800-1200 m, which is
consistence with current mountain height of ~700-800 m considering that there is also erosion caused
by landslides.

**6 Conclusions**
Here, we report the role of a Mw 6.6 Chuetsu earthquake in the topographic evolution of the young and
low mountain region, by quantitatively comparing the pre- and post- earthquake high-resolution DEMs.
Our results show, after the Chuetsu earthquake, the slope angle, relief and roughness are coseismically
increased at the epicentral area; which is different with that occurred in the old and steep Longmen Shan





orogenic region. The co-seismically induced landslides play important role in balancing the long-term
uplift by concentrated high denudation at the uplifted area far from the surface fault traces, while
according to the 2008 Wenchuan earthquake, the co-seismic denudation show different pattern that
concentrated right at the surface rupture zones. The preserved mountain peaks are not only uplifted by
thrusting and folding but also undergone erosion caused by seismically induced landslides. Finally, we
suggest that the strong earthquakes might play different roles in topographic evolutions in low and steep
mountain regions. The findings also reveal that the differential DEM method is a powerful and robust
approach in evaluating co-seismic landslide volumes as well as quantitative geomorphic analyses.

**Acknowledgements**
We appreciate Dr. Zhou Yu for his help in processing the DEM data and valuable suggestions. The
authors want to thank Geospatial Information Authority (GSI) of Japan, National Research Institute for
Earth Science and Disaster Prevention (NIED), Japan and Geological Survey of Japan (GSJ) for sharing
the topographic, landslide inventory and geological data. This work was funded by the National Key
Research and Development Program of China (2017YFC1500401), State Key Laboratory of Earthquake
Dynamics (LED2014A03) and National Natural Science Foundation of China

309 (41472201, 41761144071).






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



**Captions to figures**

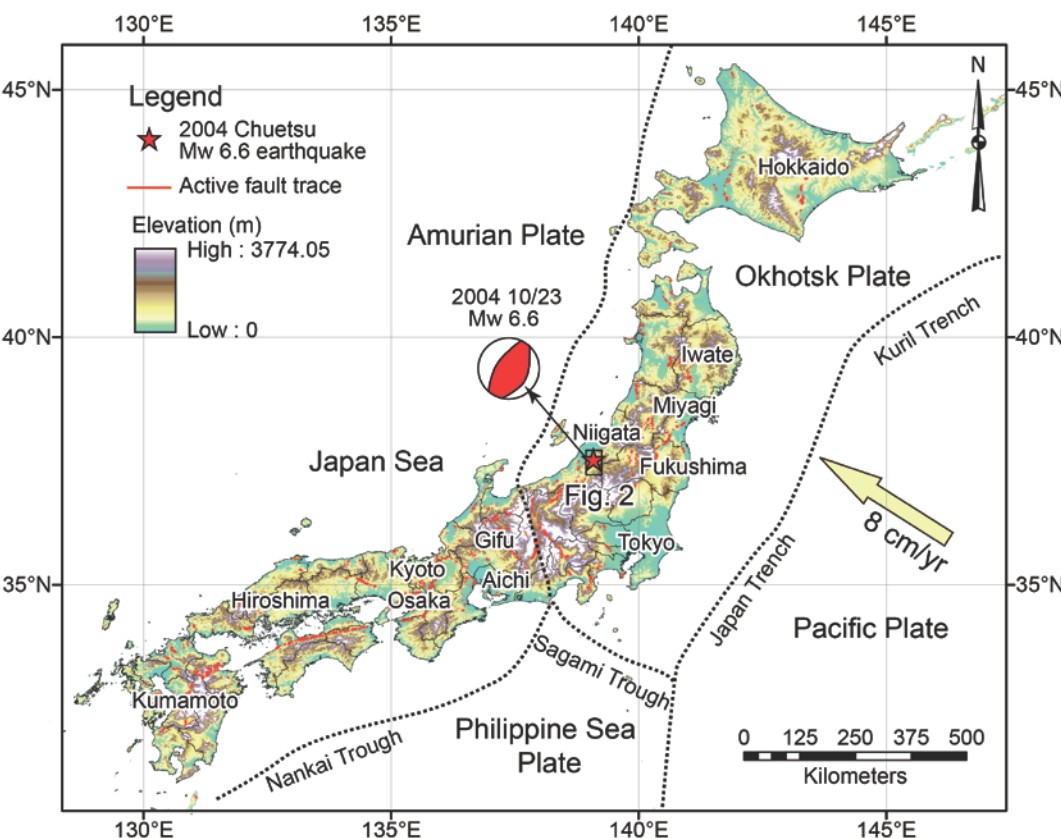

Figure 1. Plate tectonic framework and active faults in Japan. Active fault traces are from [RGFAFJ 1991; Nakata and Imaizumi, 2002]. The focal mechanism of the 2004 Mw 6.6 Chuetsu earthquake is from the Harvard Centroid Moment Tensor (CMT) catalog (http://www.globalcmt.org/CMTsearch.html). The red star indicates the epicenter of the 2004 Chuetsu earthquake. Black rectangle shows the detail location of Figure 2.



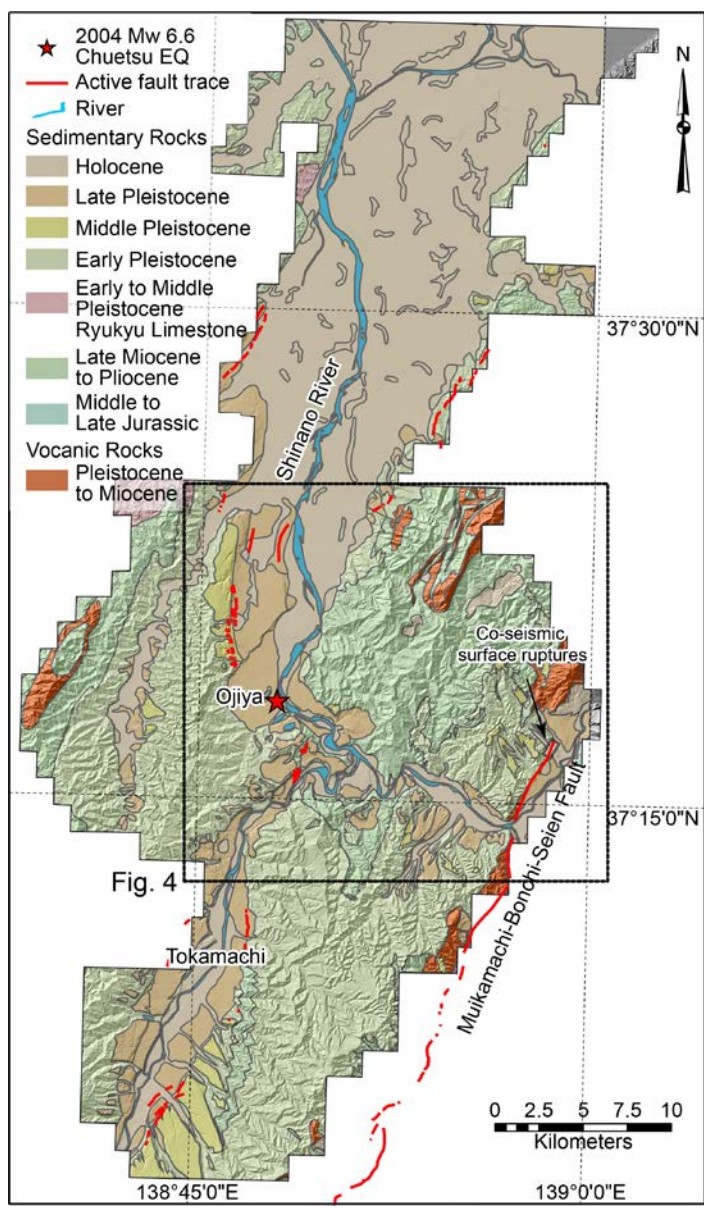

Figure 2. Geological map of the epicentral area of the 2004 Chuetsu earthquake. The red lines show the

major active faults. The small rectangle shows the location of the co-seismic surface rupture of the 2004

Chuetsu earthquake, which occurred on the northward extending of the Muikamachi fault zone

(Maruyama et al., 2005).





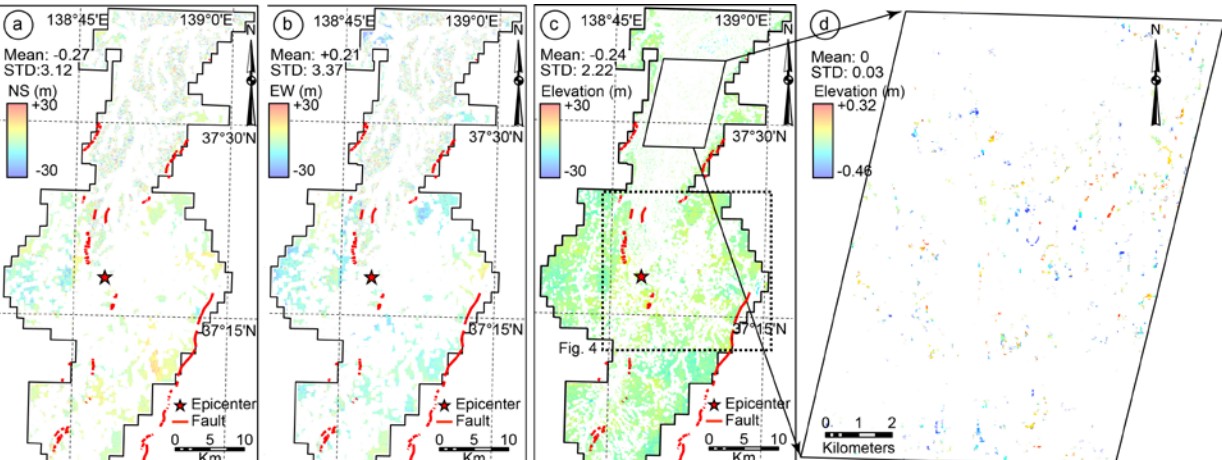

Figure 3. The horizontal differences between pre- and post-earthquake DEMs at NS direction (a), EW direction (b), the vertical deformations obtained by subtracting the pre-earthquake DEM from the post-earthquake DEM. (c), and the vertical deformations at the flat region far from the epicentral area (d). The flat region far from the epicentral area should be of no vertical deformation, which represents the accuracy of the differential DEM in this study. The dashed rectangle shows the location of Figure 4. The mean vertical deformation of the whole region is -0.24 m with standard deviation (STD) of 2.22 m. The white background indicates value of zero. The color scale is shown in min-max.





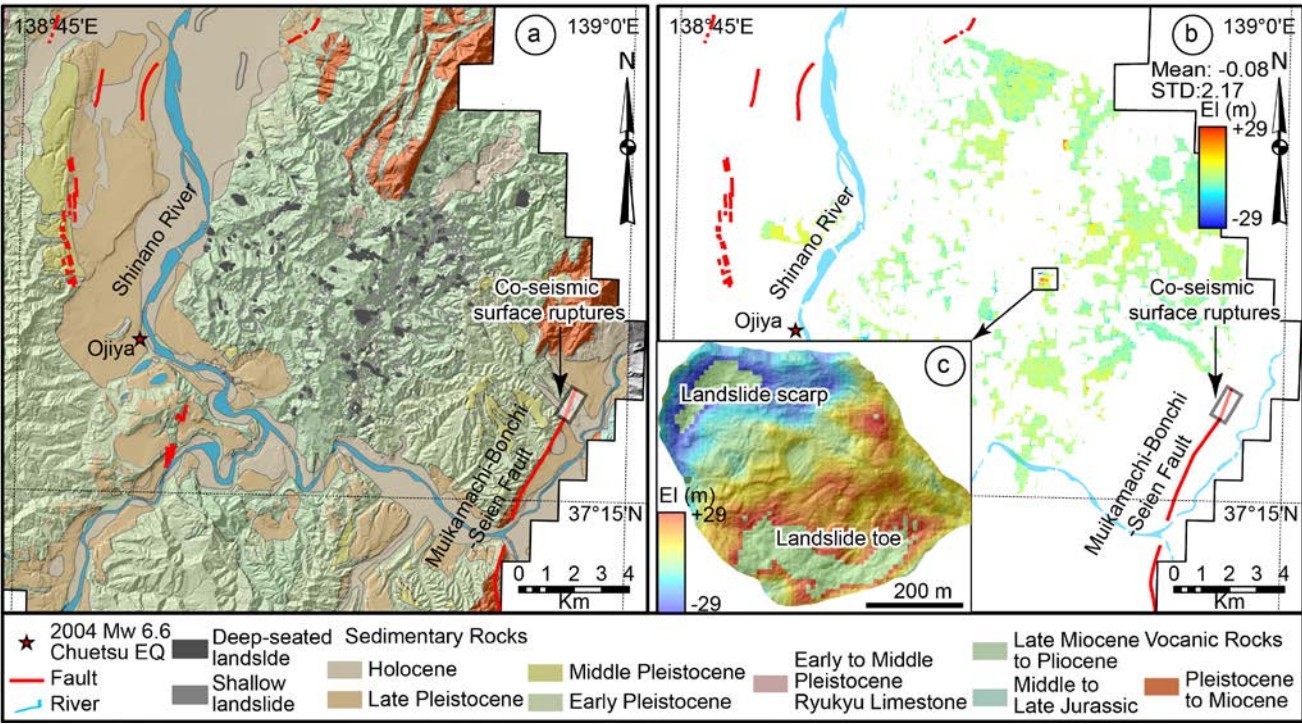

Figure 4. The geological map and distribution of co-seismic landslides (a) and vertical deformations at
the epicentral area (b), the inset map shows the largest co-seismic landslide (c). The dark gray polygons
show the deep-seated landslides and the light gray polygons show the shallow landslides. The landslide
inventory map is from the National Research Institute for Earth Science and Disaster Prevention
(NIED), Japan. The mean vertical deformation of the landslide region is -0.08 m with STD of 2.17 m.
The white background indicates value of zero. The color scale is shown in min-max.



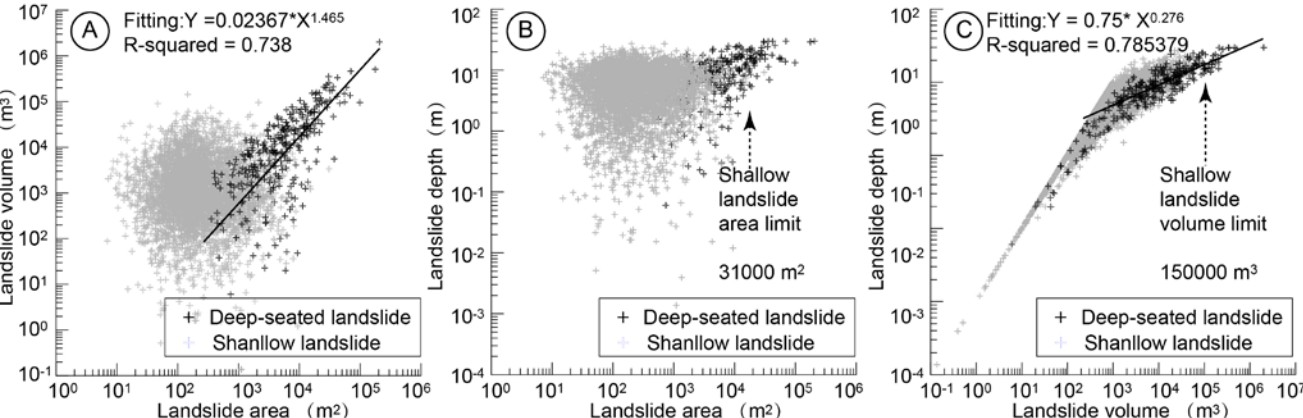

Figure 5. The relationship of landslide volume, landslide area and landslide depth. Landslide area and volume (a), Landslide depth and area (b), Landslide depth and landslide volume (c) obtained from pre- and post-earthquake DEMs.





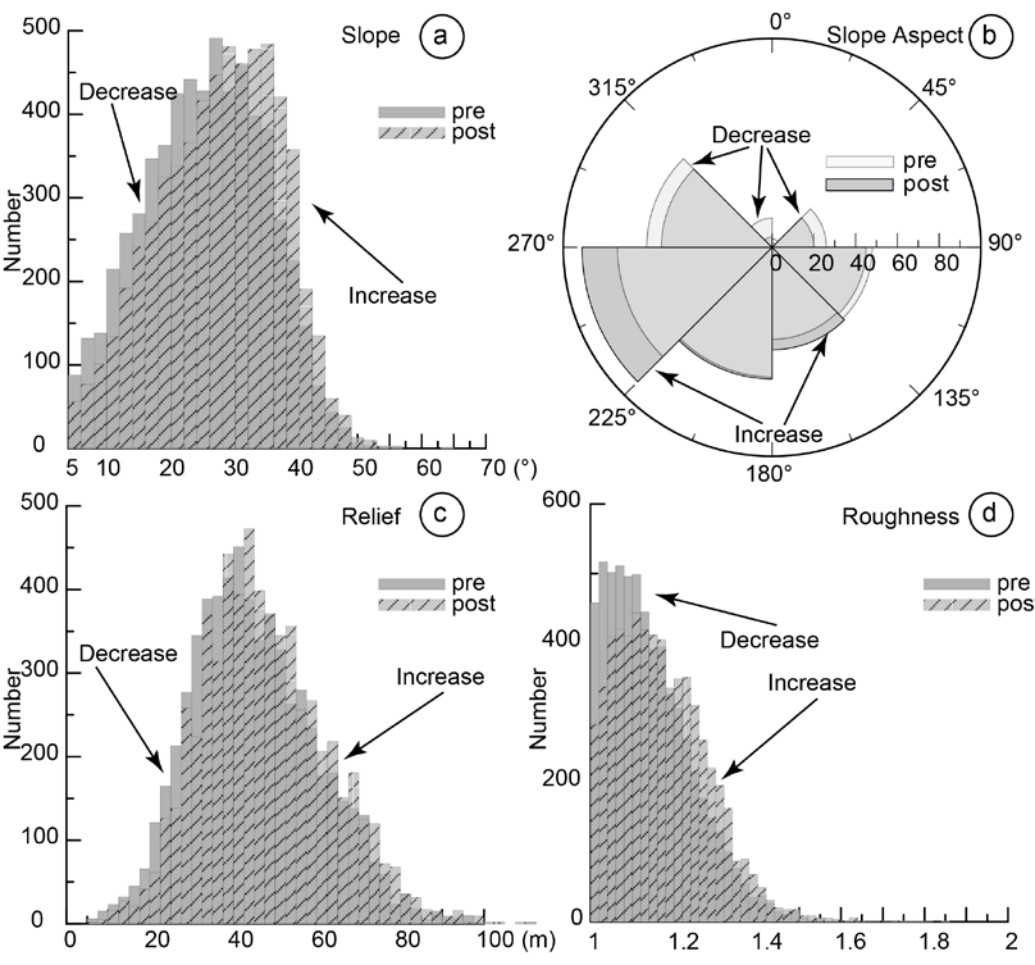


Figure 6. Statistical comparison of the pre- and post- earthquake slope angle (a), slope aspect (b), relief
(c) and roughness (d).



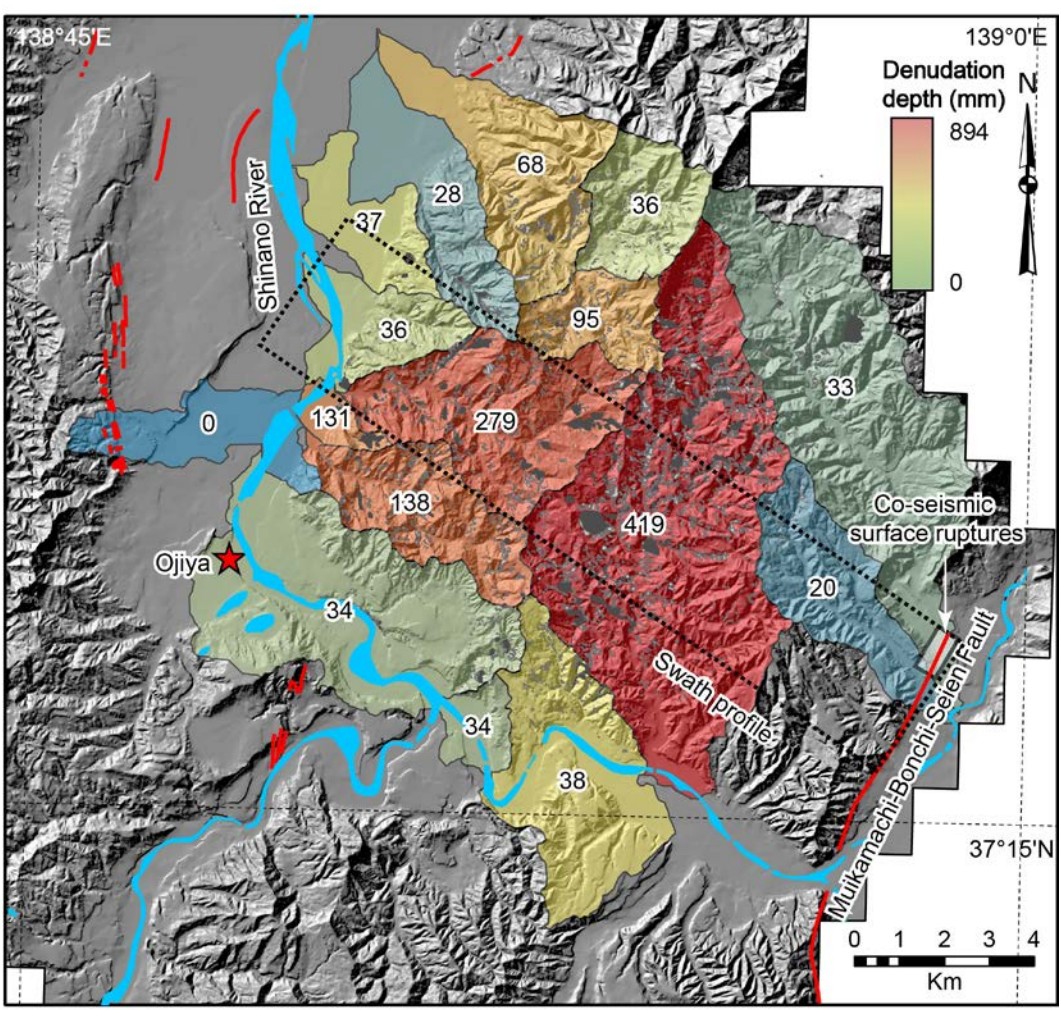


Figure 7. The distribution of catchment-scale average denudation. The denudation depths are obtained

by averaging the total landslide volume by the catchment area within each catchment.





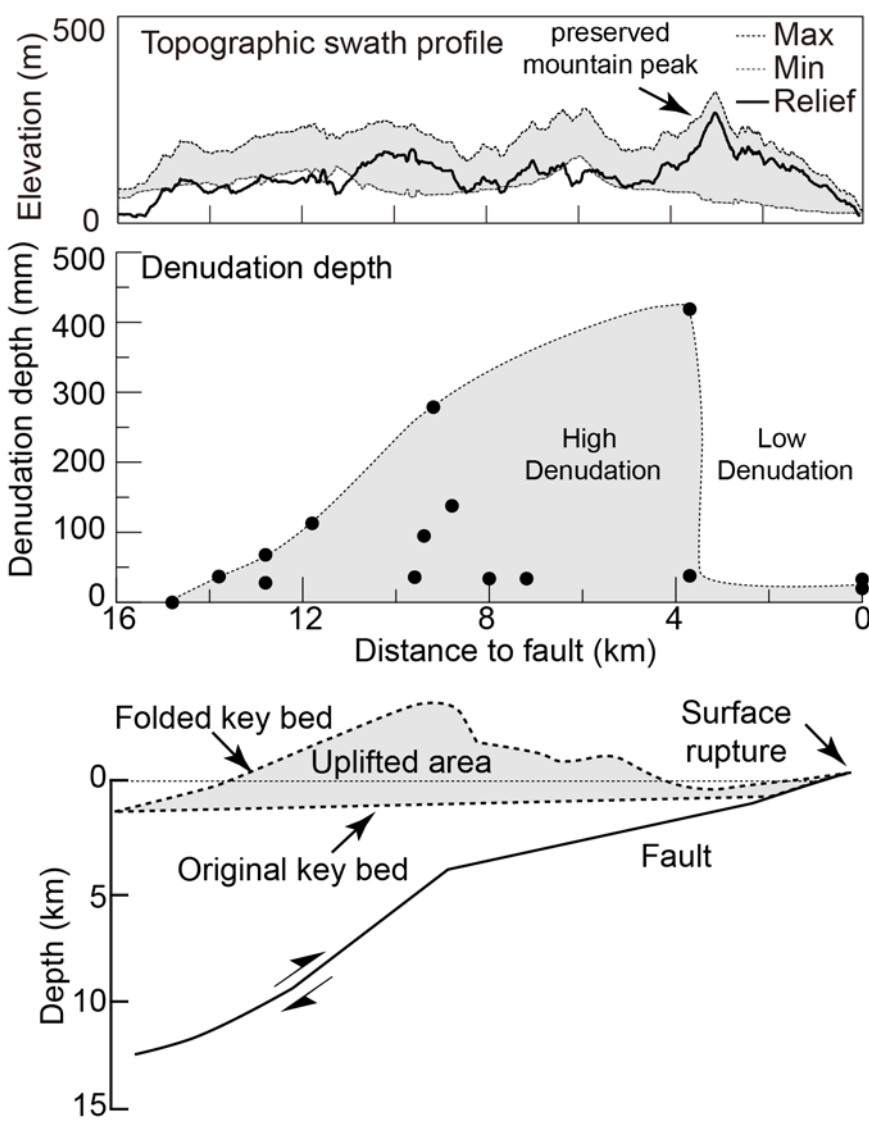

Figure 8. The topography swath profile, denudation depth and deformation pattern associated with

fault-related folding of the Muikamachi fault. The deformation pattern was modified from (Kato et al.,

2005; Kato et al., 2006; Okamura et al., 2007).