# Peer review of "Topographic changes due to the 2004 Chuetsu thrusting earthquake in low mountain region"

_Solid Earth, 2019_

## Referee Comment (RC1) · Anonymous Referee #1 · 6 Feb 2019

Review of Ren et al., on the mass balance of the Niigata earthquake

The authors present a comparison of a pre earthquake (10m, from pairs of stereo images) and post-earthquake (2m LIDAR) digital elevation models. By making the difference after co-registrating them they claim to obtain landslide volume and thus to be able to better constrain the relation between coseismic uplift and coseismic erosion in a low relief region. Although this may be a relevant topic, the study currently contains major errors that plague almost all results.

Major comments The study is poorly written and many sentences are unclear, but the key issues are methodological errors. Almost at the end of the manuscript the authors admit that the pre DEM contain vegetation and not the post-DEM. In effect this means that where a landslide occur, its estimated depth will be  $D^*=D+Ht$ , with D its real depth

and Ht the tree height. Of course for landslide on grass land (Ht«1) or very deep slides (say D~20-30) Ht (typically 3-10m) may not matter. But in places with forest yes. The authors dismiss it because they find a difference of about 0 in a sub area of their study. They simply forgot to say that this zone is an alluvial plain covered mainly by towns and field, thus with Ht~0 in most places. Any satellite image demonstrate this see the 3 figures of this review.

In contrast they "surprisingly" report (in Fig 5 absolutely no comment in the text) that almost all small landslides (>1000m2) are 5 to 10m deep when they should be around 0.5-2m typically (Larsen et al 2010). This strongly suggest a vast majority of measurement is tracking Ht not D. As a result almost all result and consequent discussion are flawed and not worth further consideration until the author make an in-depth analysis of where canopy effect may play, how much, and what are the resulting uncertainties on individual landslide and estimated erosion.

Some additional Line by Line comments Abstract: Several unclear sentences

L41: This sentence is very vague and confusing. The idea that earthquake can contribute to mean topographic base level increase, as to the formation of relief is pretty old (see King et al., 1988, Avouac 2007). We also know other tectonic processes than earthquake redistribute mass and affetc topography (e.g., interseismic processes). These facts are not very well introduced by the authors overall in the whole introduction.

L46-48 : Several of this reference are erroneous : some work do not relate earthquake to topography : e.g.; Montgomery and Larsen 2012 Hovius 2011 is about Taiwan , not the LongMenShan

L49: confusing wording: demonstrated that Ldsl are tought to ?

L51 I would suggest to specify first the location : Recent study in the arid foothills of Peru

L63 end of the sentence unclear

SED
L64 what mean totally different volumes ? Which different methods ? Again wrong reference Marc 2015 does not present anything about coseismic landslide volume.

L112 : Change the wording

L118 : If pre eartquake DEM is from stereo pairs, the height of vegetation will be included. So how do you account for it? Is there a correction on the pre-DEM ? Then it need to be explain and its uncertainties described. Or is the Lidar giving you the post-DEM with elevation including tree height ? But in this case many landslide "depth" will be driven by tree height. For me this is a likely explanation of why most of the small landslides (10-1000m2) depth is between 5-10m in Fig 5. If you look at the global database of landslide (Larsen et al., 2010) in the size range the mean depth should be around 1m (with significant scatter).

L207 : Where can we see that ?The authors need to support this claim with a supplementary figure at least.

L208: I disagree with its claim. Where there is no landslides the precision of the difference DEM may be high (low noise level) but this noise level may very well change between the landslide zone (with steep topography even if not very tall) and the valley to the North. Additionally the author compare the biggest landslides to the mean noise...They should compare to mean landslide depth.

L212 : How were shallow and deep-seated landslide classified ? Clearly I would not call deep seated a slide with a 1-2 m depth, while some of them have 10cm... And 10m is not shallow and except in some place most likely much deeper than the soil layer.

L234-238 : Here the author acknowledge, very late, the problem of canopy (this should be done in the method !). And then dismiss it on the argument that no systematic error are observed and that low elevation difference exist in the "blank area". Well the reason is simple enough : in the carefully selected area of fig 4b, there is only city,
agricultural fields and river flood plain. SO very limited high ranging vegetation, and if the images were taken when fields were denuded it would make sense to obtain overall no elevation change.

Authors figures Fig 1-4 Overall the author show lithological maps everywhere that are almost not discussed. IN contrast a map differentiating agricultural lands, forest, grass land, and shrub or medium height vegetation would be much more useful and a potential place to start to evaluate methodological error related to canopy.

Fig 1: could be an inset.

Fig 3C : The color scale need to be re adjusted to something like +10 / -10 Same for Fig 4, this stretch hides all details and just show the biggest slides.

Fig 5: We need to see the uncertainties on the parameters of the V-A relationships and the associated confidence interval on the plot (along the fitted lines).

In any case the biggest issue is that the trend of size with depth do not exist for "shallow" landslides. They tend to be randomly distributed around 10m, that is most likely a methodological error, not acknowledged nor discussed by the authors.

Fig 8B : No idea what the points are or the shade line and how they are drawn...

References used in the review: Avouac, J.-P. (2007), 6.09 Dynamic processes in extensional and compressional settings—Mountain building: From earthquakes to geological deformation, in Treatise on Geophysics, edited by E.-C. G. Schubert, pp. 377–439, Elsevier, Amsterdam

King, G. C. P., R. S. Stein, and J. B. Rundle (1988), The growth of geological structures by repeated earthquakes 1. Conceptual framework, J. Geophys. Res., 93(B11), 13,307–13,318, doi:10.1029/JB093iB11p13307.

Larsen, I. J., Montgomery, D. R., & Korup, O. (2010). Landslide erosion controlled by hillslope material. Nature Geoscience, 3(4), 247.

SED

---

## Referee Comment (RC2) · Anonymous Referee #1 · 3 Apr 2019

Dear Editor,

I have read the rebuttal letter of the authors. I see the authors have improved the english and clarified a number of minor points.

However they have refused to include any substantial change or additional discussion on their methods and results about landslide volume. I am not convinced by the different arguments of the authors and still consider that the study cannot be published without a rigorous assessment of the methods uncertainties. The authors suggest some of my suggestions are impossible but in the contrary I think the authors have access to the data and just need to perform a series of test and analysis that should not take longer than a few weeks.

[Figure]

I discuss below the points raised with the authors with which I am still disagreeing. I have also tried to clarify my current issues and suggestion with the current analysis of the authors.

Major comment

The author argue canopy effects is not important but I do not see their arguments as very convincing. On my opinion they present several conflicting or poorly supported arguments (I briefly summarize them before commenting on them): > 1/ The GSI has GCP ground control point. Ok but unless the author show a map of them and show that they are included in the occasional forested land I do not think it exclude that the digital surface will go over forested lands.

2/ Negative and positive part should not be comparable (difference by tree removal should always be much bigger than positive deposits ). I agree and this is a good approach to validate their methods and data. However I do not see a good support in the current revised paper. I currently cannot understand what was done with the Suppl. Table : There are ∼300 entries with area (of the landslide ?) and a fraction of them have indicated a Min Max Mean Std and Average Error. Presumably the result of the DEM difference. This table is not describe din the revision. I do not know why most data are not reported, or how was obtained the average error. I do not understand why some landslides have a MEAN change large and positive (5m or more) that would suggest the deposit is much larger and thicker than the eroded area, that does not make sense from a mass balance point of view. . .

3/ The absolute value of landslide volume is not the main contribution. » I think this is clearly contradicted by the whole abstract that can be summarized as: " Landslide volume matter to understand the impact of EQ on topography. It is hard to estimate. We propose a new method. We use landslide volume to obtain denudation and thus EQ mass balance and discuss it". I think the authors cannot escape a more detailed discussion on their landslide volume estimate. If biased the whole tectonic discussion

would be too.

4/ This argument is mixing several points : forest is not everywhere ; Not all landslides are on forested slopes ; Lidar may be blocked by forest when very dense; I agree that forest is not everywhere, but it does not remove the need to examine quantitatively the errors caused by the presence of forest. Again, the authors have aerial imagery available, it would be easy to select a number of forested zone where landslide have occurred and another set of landslides occurring on grassland. Then plot the 2 different dataset (could be just colored point in Fig 5), and discuss possible difference bias. It would be especially important for the medium size landslides.

If Lidar is also blocked by forest, it would explain they find overall similar DEM where landslide did not occurred but the difference between Pre and Post landsliding would still be dominated by the removal of the tree so this point does not answer my worries (although the author should check and write in the methods whether of nor the LIDAR is expected to have passed through the canopy.

The authors conclude that : "At least, our results are much more reliable than the volume information from scaling laws; we use real pre- and post-earthquake data." This is currently not supported. First, using "data" rather than a model does not mean that there cannot be biased in the data. Then the scaling laws are also based on real (field) measurements, from around the world, and show a strong convergence (with many different type of landslides through out the world having the same trend (Guzzeti et al., 2009, Larsen et al., 2010). Of course there is some statistical noise, but the trend and order of magnitude of the parameters have been validated many times. The data of the authors (for their so called shallow landslide) do not follow the expected trend and that is worrying.

Respnse to some Line By Line Comments L118/ L217 : The author reply that their landslide depth is validated in Fig 4B and 4C.

This cannot be currently judged. As I said in my review, it is unsurprising that the

biggest landslide show reasonable volume change (erosion/deposit) as its depth is big relative to tree height. Obviously as seenable in Fig 5, many slides have reasonable depth and many have very ssuspicious depth. One working example cannot help to judge the whole dataset.

L208: I previously suggested that some more validation should be performed abou the uncertainties and bias of the DEM difference. The authors reply to me that study using scaling laws also have uncertainties but have been published, and somehow suggest the best analysis have already been done and that more testing would be "mission impossible". » 1/ errors in other papers do not justify to reduce the quality of your own paper. And there are a number of evidence from the manuscript figure that cast serious doubt on the volume obtained in this study, explaining why I am currently doubting the overall discussion and conclusions. 2/ My point still hold and I make precise suggestion that are totally doable (the authors have all the data) : Take any imagery distinguishing forest and land, subset landslide in forested areas and in grass lands, analyze and plot the negative and positive volume in these 2 subsets. Compare the Area-Volume-Depth in these 2 zones. This would be a first easy step to assess the impact of canopy.

L212 : Definition of the deep-seated landslide. The authors did not respond to my point. I am sorry I have not mentioned my point is based on Fig 5B : IN that we see clearly that about 30 "Deep Seated" slides (black crosses) have depth below 1-2 m. The smallest with 10cm. This is average depth according to the authors and it is preety worrying. Similarly the "shallow" slides according to the independent inventories (from satellited and field if I understand from the reply letter) have very frequently ∼10m or more depth. . . This is also suggesting there are problems either in the DEM volume estimate or in the landslide classification or in both.

Fig3C : I do not think that the authors reply makes sense: Currently we see nothing else than the big landslides (i.e. a couple of zones) and then many patches around 0. If the authors use a color scale saturating at -10 and 10, the shallow slides will be more visible AS WELL AS the deep ones (only they will have uniform blue and red

[Figure]

erosion/deposit saturated at -10/10). Fig 4C can still use the -30/30 scale to show the detail of course.

Anyway to answer my comments about the evaluation of landslide depth retrieval, it would be essential that a main text figure OR a supplementary figure, show a few zones (with and without forest ) with the color map of DEM changes, and the landslide polygon superposed. 2-3 Zooms containing a dozens of polygons with different size in different setting would be perfect. Again this is extremely easy to do and I do not understand why the authors refuse to show that in the supplement.

Fig 5 : Well there are hundreds of points (grey crosses) indicating shallow landslides, with area between 10 and 10,000 m2 and with depth around 10m. I just looked at the data (Of course I also do see many shallow landslides with depth around 0.5-2m, as you expect from scaling relationships).

[Figure]

---

## Author Comment (AC1) · 3 Apr 2019

Editor: Solid Earth Dear Editor in chief: I am sending herewith the revised manuscript titled on "Topographic changes due to the 2004 Chuetsu thrust earthquake in a low mountain region" for possible publication in the " Solid Earth". Thanks for your email on 6th Feb, 2019 to inform us that the Referee comment was posted. During the revision, we explained that why our results did not include canopy effects in a separate response file. Meanwhile, we provide a supplementary file to prove it. We also send our manuscript to a professional company to polish the English as the referee point out that our English is not good enough. Hope now our explanation is acceptable. This manuscript has not been previously published and is not and will not be submitted for publication elsewhere when it is in review for the " Solid Earth ".

[Figure]

Sincerely yours, Zhikun Ren

State Key Laboratory of Earthquake Dynamics Institute of Geology China Earthquake Administration No.1 Huayanli, Chaoyang district, Beijing 100029, P.O. Box 9803, China Tel. & Fax:(+86)-10-62009085 Email: rzk@ies.ac.cn lzkren@gmail.com

Reply and correspondence to the Reviewer's comments and suggestions: I am grateful to the reviewer for his/her positive and critical comments and suggestions. The following main revisions and answers are made in reply to the reviewer's general comments and suggestions; these changes have also been made in the text.

Major comments: The study is poorly written and many sentences are unclear, but the key issues are methodological errors. Almost at the end of the manuscript the authors admit that the pre DEM contain vegetation and not the post-DEM. In effect this means that where a landslide occur, its estimated depth will be $D^*=D+H_t$ , with D its real depth and $H_t$ the tree height. Of course for landslide on grass land ($H_t\approx1$) or very deep slides (say $D\approx20$-30) $H_t$ (typically 3-10m) may not matter. But in places with forest yes. The authors dismiss it because they find a difference of about 0 in a sub area of their study. They simply forgot to say that this zone is an alluvial plain covered mainly by towns and field, thus with $H_t\approx0$ in most places. Any satellite image demonstrate this see the 3 figures of this review. In contrast they "surprisingly" report (in Fig 5 absolutely no comment in the text) that almost all small landslides (>1000m2) are 5 to 10m deep when they should be around 0.5-2m typically (Larsen et al 2010). This strongly suggest a vast majority of measurement is tracking $H_t$ not D. As a result almost all result and consequent discussion are flawed and not worth further consideration until the author make an in-depth analysis of where canopy effect may play, how much, and what are the resulting uncertainties on individual landslide and estimated erosion.

Response:First, the canopy effect is not as serious as the reviewer suggested. The pre-earthquake DEM is not generated simply from the stereo pair of images but also from the field survey GCP. It is not produced by us but is downloaded from the Geospatial Information Authority (GSI) of Japan (Freely available at http://fgd.gsi.go.jp/download); thus, this DEM is already calibrated based on field survey data in order to show the bare earth surface DEMïijĹbut not 100%ïijĽ. Second, if the results include the canopy part, then the positive and negative values of the DEM difference should not be comparable. The negative values should be much larger than the positive values because the pre-earthquake DEM is assumed to include the forest, i.e., the Ht. However, according to our results, the results clearly show the landslide scarp and toe; hence, positive and negative values are comparable. If there are large errors caused by the canopy, then the values should be mostly large and negative, or at least they should be systematically negative with respect to the Ht value (typically 3-10 m) as the reviewer commented. In the revised manuscript, we provide a supplementary file that indicates the mean error of the difference elevation by summing the positive and negative values, which indicates that the mean difference in the landslide region is mostly on the millimetre to centimetre scales (Please refer to the last column of Supplementary Table 1). This result indicates that the average value is almost zero, which implies that the negative and positive values are comparable; hence, there are no obvious canopy effects. Third, the absolute values of the landslide volumes are not the main contribution of our research. We finally propose a distribution pattern for the erodible material caused by the landslides. By comparison with the geological model, we discuss the role of the earthquake in topographic evolution. Fourth, with respect to the images mentioned in the reviewer comments, even if there is forest cover, there are also many bare earth surfaces, as shown in Figure 1, such as roads and farmlands (with almost no trees at all), and only part of the mountain top (it should be less than 50% from Figures 1 and 3) is covered by forest. The southernmost part of Figure 1 shows the coverage of forest, which is difficult to calibrate to obtain bare earth DEM. However, the mountain top is usually not vulnerable to landslides, and landslides usually occur on mountain slopes. Hence, our results do not calculate the whole region after differential DEM, and we mainly focus on the reliable results (the most seriously forested region is almost all removed and show much less change in elevation), as

shown in Figure 4B. Fifth, theoretically, if as the reviewer suggested, the canopy effect is so serious, then even LiDAR could obtain only the surface of the top of canopy, then there would not be a problem using pre- and post-earthquake DEMs to obtain the landslide volume and topographic changes.

Hence, we think that the reviewer's comments about the canopy effect do not address a main problem in our manuscript. At least, our results are much more reliable than the volume information from scaling laws; we use real pre- and post-earthquake data.

Detail CommentsïijŽ L41: This sentence is very vague and confusing. The idea that earthquake can contribute to mean topographic base level increase, as to the formation of relief is pretty old (see King et al., 1988, Avouac 2007 ). We also know other tectonic processes than earthquake redistribute mass and affetc topography (e.g., interseismic processes). These facts are not very well introduced by the authors overall in the whole introduction.

ReplyïijŽThis sentence explains our main idea about the co-seismic effects caused by strong earthquakes in topographic evolution. We cannot consider the old view of earthquakes contributing only to the topographic base level increase (this concept is only partially correct); actually, this view is not true because we use high-resolution pre- and post-earthquake DEMs, InSAR studies, GPS observations, etc. Only a small region along the co-seismic surface rupture zone is uplifted, and only when a thrust earthquake occurs. If the fault is normal, even the region around the co-seismic surface rupture zone is depressed. Regarding a strike-slip fault, if it causes many co-seismic landslides, it also mainly generates an elevation decrease rather than an increase. The study by McPhilips, "the Millennial-scale record of landslides in the Andes consistent with earthquake trigger" also shows the role of earthquakes in topographic evolution. We actually do not want to confuse the readers about some old incomplete views of the role of earthquakes in topographic evolution; therefore, we do not include these quite old references.

L46 – 48 : Several of this reference are erroneous : some work do not relate earthquake to topography : e.g.; Montgomery and Larsen 2012 Hovius 2011 is about Taiwan , not the LongMenShan Reply. We apologize for the wrong insertion of references. We have modified the citation.

L49: confusing wording: demonstrated that Ldsl are tought to ? Reply: Thank you for the reviewer's valuable comments. The sentence has been revised to clarify what we would like to express to the readers. "Previous studies have demonstrated that landslides limit the slope"

L51 I would suggest to specify first the location : Recent study in the arid foothills of Peru Reply: Thank you for the reviewer's valuable comment. The sentence is revised by showing the location as the reviewer suggested. "Recent study in the arid foothills of Peru found that erosion caused by landslides did not change much in response to climatic changes;"

L63 end of the sentence unclear Reply: Thank you for the reviewer's valuable comments. This sentence demonstrates the use of scaling laws to obtain co-seismic landslides; there are large uncertainties. The sentence is revised to "However, using scaling laws to obtain the co-seismic landslide volumes has large uncertainties in different regions."

L64 what mean totally different volumes ? Which different methods ? Again wrong reference Marc 2015 does not present anything about coseismic landslide volume. Reply: Thank you for the reviewer's valuable comments. The sentence is revised as follows: "Different co-seismic landslide volume results have been reported for the 2008 Wenchuan earthquake"

L112 : Change the wording Reply: Thank you for the reviewer's comments. The sentence is revised as follows: "Hence, the topographic evolution in the epicentral area should be closely related to the co-seismic landslides caused by strong earthquakes."

L118 : If pre eartquake DEM is from stereo pairs, the height of vegetation will be included. So how do you account for it? Is there a correction on the pre-DEM ? Then it need to be explain and its uncertainties described. Or is the Lidar giving you the post-DEM with elevation including tree height ? But in this case many landslide "depth" will be driven by tree height. For me this is a likely explanation of why most of the small landslides (10-1000m2) depth is between 5-10m in Fig 5. If you look at the global database of landslide (Larsen et al., 2010) in ths size range the mean depth should be around 1m (with significant scatter). Reply: Please refer to the reply to the main comments above. First, the pre-earthquake DEM is corrected. Meanwhile, there is actually a large area of bare earth in the epicentral area that is not densely forested. Even if we assume that there are serious canopy effects, they would not be consistent with our results from the differential DEM (negative values should not be comparable with positive values). It is also possible that the LiDAR data, including the forest canopy, are too dense. Overall, we believe our DEM shows real results for the topographic difference, as indicated by the landslide scarp and landslide toe area in one landslide in Figure 4c. Theoretically, the global landslide dataset could not show the pattern of co-seismic landslides at all; it is of different scale. Meanwhile, the landslide inventory data are from the formally available data (compiled not by one researcher but by a group of Japanese researchers, which also includes much field work; we believe the results are reliable because the deep-seated and shallow landslides are from the combination of remote sensing image interpretation and field validation).

L207 : Where can we see that ?The authors need to support this claim with a sup-plementary figure at least. Reply: Thank you for the reviewer's comments. Figure 4b shows the reliable results for the overall topographic changes. By applying the land-slide inventory polygon, we clearly show the landslide scarp and landslide toe area in Figure 4c.

L208: I disagree with its claim. Where there is no landslides the precision of the difference DEM may be high (low noise level) but this noise level may very well change

between the landslide zone (with steep topography even if not very tall) and the valley to the North. Additionally the author compare the biggest landslides to the mean noise. . .They should compare to mean landslide depth. Reply: Thank you for the reviewer's comments. There are indeed different precisions according to the DEMs. The studies using scaling laws to derive landslide volume include many large errors, but many papers have been published. We are using higher-resolution data and results, which is clearly shown in the manuscript, but the reviewer seems not to believe it. We have done our best to explain our study and try to show more reliable results to the reader. However, there are also many "mission impossible" tasks in scientific research.

L212 : How were shallow and deep-seated landslide classified ? Clearly I would not call deep seated a slide with a 1-2 m depth, while some of them have 10cm. . . And 10m is not shallow and except in some place most likely much deeper than the soil layer. Reply: Thank you for the reviewer's comments. I am not sure where the reviewer found that deep-seated landslides are 10 cm. Our results show the overall topographic changes. Even if the reviewer's point is true, it is possible that a deep-seated landslide may have some less deformed shallow parts at the edges.

L234-238 : Here the author acknowledge, very late, the problem of canopy (this should be done in the method !). And then dismiss it on the argument that no systematic error are observed and that low elevation difference exist in the "blank area". Well the reason is simple enough : in the carefully selected area of fig 4b, there is only city, agricultural fields and river flood plain. SO very limited high ranging vegetation, and if the images were taken when fields were denuded it would make sense to obtain overall no elevation change. Reply: Thank you for the reviewer's comments. However, the sequence of the figures should match the contents of the manuscript; from the overall structure of the manuscript, we could not first show the results of Figure 4 and then explain the details in the method section. We need to first show Figure 3. If we included this part in the method, maybe other reviewers would comment that the methods and results are mixed. We hope that the reviewer can understand.

Authors figures Fig 1-4 Overall the author show lithological maps everywhere that are almost not discussed. IN contrast a map differentiating agricultural lands, forest , grass land, and shrub or medium height vegetation would be much more useful and a potential place to start to evauate methodological error related to canopy.

Fig 1: could be an inset. Reply: Thank you for the reviewer's comments. This figure shows the overall tectonic setting of the Chuetsu area within a larger geological context to allow readers to better understand the background of mountain building in this region. Meanwhile, the DEM data covering Japan also show the orogenic features and mountains clearly. We actually tried to show this information as an inset figure, but a very small inset figure could not show the geological background clearly.

Fig 3C : The color scale need to be re adjusted to something like +10 / -10 Same for Fig 4, this stretch hides all details and just show the biggest slides. Reply: Thank you for the reviewer's comments. However, we do not agree with the reviewer. This disagreement is because there are deep-seated landslides, very shallow landslides and non-deformed regions. The scale should not be stretched to show false information by adjusting the real elevation change to +10 to -10. This action would mean that we removed other deformations between +30 to +20 and -30 to -20.

Fig 5: We need to see the uncertainties on the parameters of the V-A relationships and the associated confidence interval on the plot (along the fitted lines). In any case the biggest issue is that the trend of size with depth do not exist for "shallow" landslides. They tend to be randomly distributed around 10m, that is most likely a methodological error, not acknowledged nor discussed by the authors. Reply: Thank you for the reviewer's comments. We do not understand the point of this comment. The figure is in a log plot because the scatter of landslide depths is so large, but how could the figure be interpreted by the reviewer to indicate ∼10 m?

Fig 8B : No idea what the points are or the shade line and how they are drawn... Reply: Thank you for the reviewer's comments. The points show the average denudation

depths of the catchments. This figure is just a very simple plot using X (distance to the fault) and Y values (denudation depth of the catchment).

Please also note the supplement to this comment:
https://www.solid-earth-discuss.net/se-2019-3/se-2019-3-AC1-supplement.zip
* * *

---

## Author Comment (AC2) · 4 Apr 2019

Thanks for the referee's quick review. We provide a supplementary file to show the positive and negative values to be comparable to prove our results are reliable. In our reply we demonstrated major reason why canopy should not be included in the differential results of the landslide volume. The strongest evidence is the data in the supplementary file, the "0" value means the negative and positive value is almost the same (by adding them together its 0). Thanks for your comments.

---

## Referee Comment (RC3) · Anonymous Referee #2 · 17 May 2019

There are grounds for concern over the handling of canopy effects in this paper. I come to this review late in the process, and can see the notes and discussion between the first reviewer and the authors, which are very useful in providing these comments.

In essence, the discussion of canopy effects should be more central to the paper and not have been tacked on as afterthoughts and reactions to the first review.

So, while the subject is interesting and a lot of work has gone in to the paper, it does not look robust enough for publication in its present form. It would make sense for the authors to have a complete re-think of their analysis, before considering what to do next. Wherever the paper is taken, it is going to meet the same comments at the review stage.

---

## Author Comment (AC3) · 30 May 2019

Reply and correspondence to the Reviewer's comments and suggestions: I am grateful to the reviewer for his/her positive and critical comments and suggestions. The following main revisions and answers are made in reply to the reviewer's general comments and suggestions; these changes have also been made in the text. Review 1ïijŽ Major comments: The study is poorly written and many sentences are unclear, but the key issues are methodological errors. Almost at the end of the manuscript the authors admit that the pre DEM contain vegetation and not the post-DEM. In effect this means that where a landslide occur, its estimated depth will be $D^*=D+Ht$ , with D its real depth and Ht the tree height. Of course for landslide on grass land ($Ht\hat{A}\acute{n}1$) or very deep slides (say $D\hat{a}\acute{L}ij20\text{-}30$) Ht (typically 3-10m) may not matter. But in places with forest yes. The

authors dismiss it because they find a difference of about 0 in a sub area of their study. They simply forgot to say that this zone is an alluvial plain covered mainly by towns and field, thus with HtâĹij0 in most places. Any satellite image demonstrate this see the 3 figures of this review. In contrast they "surprisingly" report (in Fig 5 absolutely no comment in the text) that almost all small landslides (>1000m2) are 5 to 10m deep when they should be around 0.5-2m typically (Larsen et al 2010). This strongly suggest a vast majority of measurement is tracking Ht not D. As a result almost all result and consequent discussion are flawed and not worth further consideration until the author make an in-depth analysis of where canopy effect may play, how much, and what are the resulting uncertainties on individual landslide and estimated erosion.

Responseïij Ž First, the canopy effect is not as serious as the reviewer suggested. The pre-earthquake DEM is not generated simply from the stereo pair of images but also from the field survey GCP. It is not produced by us but is downloaded from the Geospatial Information Authority (GSI) of Japan (Freely available at http://fgd.gsi.go.jp/download); thus, this DEM is already calibrated based on field survey data in order to show the bare earth surface DEMïijĹbut not 100%ïijĽ. Second, if the results include the canopy part, then the positive and negative values of the DEM difference should not be comparable. The negative values should be much larger than the positive values because the pre-earthquake DEM is assumed to include the forest, i.e., the Ht. However, according to our results, the results clearly show the landslide scarp and toe; hence, positive and negative values are comparable. If there are large errors caused by the canopy, then the values should be mostly large and negative, or at least they should be systematically negative with respect to the Ht value (typically 3-10 m) as the reviewer commented. In the revised manuscript, we provide a supplementary file that indicates the mean error of the difference elevation by summing the positive and negative values, which indicates that the mean difference in the landslide region is mostly on the millimetre to centimetre scales (Please refer to the last column of Supplementary Table 1). This result indicates that the average value is almost zero, which implies that the negative and positive values are comparable; hence, there are

no obvious canopy effects. Third, the absolute values of the landslide volumes are not the main contribution of our research. We finally propose a distribution pattern for the erodible material caused by the landslides. By comparison with the geological model, we discuss the role of the earthquake in topographic evolution. Fourth, with respect to the images mentioned in the reviewer comments, even if there is forest cover, there are also many bare earth surfaces, as shown in Figure 1, such as roads and farmlands (with almost no trees at all), and only part of the mountain top (it should be less than 50% from Figures 1 and 3) is covered by forest. The southernmost part of Figure 1 shows the coverage of forest, which is difficult to calibrate to obtain bare earth DEM. However, the mountain top is usually not vulnerable to landslides, and landslides usually occur on mountain slopes. Hence, our results do not calculate the whole region after differential DEM, and we mainly focus on the reliable results (the most seriously forested region is almost all removed and show much less change in elevation), as shown in Figure 4B. Fifth, theoretically, if as the reviewer suggested, the canopy effect is so serious, then even LiDAR could obtain only the surface of the top of canopy, then there would not be a problem using pre- and post-earthquake DEMs to obtain the landslide volume and topographic changes.

Hence, we think that the reviewer's comments about the canopy effect do not address a main problem in our manuscript. At least, our results are much more reliable than the volume information from scaling laws; we use real pre- and post-earthquake data.

Detail CommentsïijŽ L41: This sentence is very vague and confusing. The idea that earthquake can contribute to mean topographic base level increase, as to the formation of relief is pretty old (see King et al., 1988, Avouac 2007 ). We also know other tectonic processes than earthquake redistribute mass and affetc topography (e.g., interseismic processes). These facts are not very well introduced by the authors overall in the whole introduction.

ReplyïijŽThis sentence explains our main idea about the co-seismic effects caused by strong earthquakes in topographic evolution. We cannot consider the old view of

earthquakes contributing only to the topographic base level increase (this concept is only partially correct); actually, this view is not true because we use high-resolution pre- and post-earthquake DEMs, InSAR studies, GPS observations, etc. Only a small region along the co-seismic surface rupture zone is uplifted, and only when a thrust earthquake occurs. If the fault is normal, even the region around the co-seismic surface rupture zone is depressed. Regarding a strike-slip fault, if it causes many co-seismic landslides, it also mainly generates an elevation decrease rather than an increase. The study by McPhilips, "the Millennial-scale record of landslides in the Andes consistent with earthquake trigger" also shows the role of earthquakes in topographic evolution. We actually do not want to confuse the readers about some old incomplete views of the role of earthquakes in topographic evolution; therefore, we do not include these quite old references.

L46 – 48 : Several of this reference are erroneous : some work do not relate earthquake to topography : e.g.; Montgomery and Larsen 2012 Hovius 2011 is about Taiwan , not the LongMenShan Reply. We apologize for the wrong insertion of references. We have modified the citation.

L49: confusing wording: demonstrated that Ldsl are tought to ? Reply: Thank you for the reviewer's valuable comments. The sentence has been revised to clarify what we would like to express to the readers. "Previous studies have demonstrated that landslides limit the slope"

L51 I would suggest to specify first the location : Recent study in the arid foothills of Peru Reply: Thank you for the reviewer's valuable comment. The sentence is revised by showing the location as the reviewer suggested. "Recent study in the arid foothills of Peru found that erosion caused by landslides did not change much in response to climatic changes;"

L63 end of the sentence unclear Reply: Thank you for the reviewer's valuable comments. This sentence demonstrates the use of scaling laws to obtain co-seismic landslides; there are large uncertainties. The sentence is revised to "However, using scaling laws to obtain the co-seismic landslide volumes has large uncertainties in different regions."

L64 what mean totally different volumes ? Which different methods ? Again wrong reference Marc 2015 does not present anything about coseismic landslide volume. Reply: Thank you for the reviewer's valuable comments. The sentence is revised as follows: "Different co-seismic landslide volume results have been reported for the 2008 Wenchuan earthquake"

L112 : Change the wording Reply: Thank you for the reviewer's comments. The sentence is revised as follows: "Hence, the topographic evolution in the epicentral area should be closely related to the co-seismic landslides caused by strong earthquakes."

L118 : If pre eartquake DEM is from stereo pairs, the height of vegetation will be included. So how do you account for it? Is there a correction on the pre-DEM ? Then it need to be explain and its uncertainties described. Or is the Lidar giving you the post-DEM with elevation including tree height ? But in this case many landslide "depth" will be driven by tree height. For me this is a likely explanation of why most of the small landslides (10-1000m2) depth is between 5-10m in Fig 5. If you look at the global database of landslide (Larsen et al., 2010) in ths size range the mean depth should be around 1m (with significant scatter). Reply: Please refer to the reply to the main comments above. First, the pre-earthquake DEM is corrected. Meanwhile, there is actually a large area of bare earth in the epicentral area that is not densely forested. Even if we assume that there are serious canopy effects, they would not be consistent with our results from the differential DEM (negative values should not be comparable with positive values). It is also possible that the LiDAR data, including the forest canopy, are too dense. Overall, we believe our DEM shows real results for the topographic difference, as indicated by the landslide scarp and landslide toe area in one landslide in Figure 4c. Theoretically, the global landslide dataset could not show the pattern of co-seismic landslides at all; it is of different scale. Meanwhile, the landslide inventory data

are from the formally available data (compiled not by one researcher but by a group of Japanese researchers, which also includes much field work; we believe the results are reliable because the deep-seated and shallow landslides are from the combination of remote sensing image interpretation and field validation).

L207 : Where can we see that ?The authors need to support this claim with a supplementary figure at least. Reply: Thank you for the reviewer's comments. Figure 4b shows the reliable results for the overall topographic changes. By applying the landslide inventory polygon, we clearly show the landslide scarp and landslide toe area in Figure 4c.

L208: I disagree with its claim. Where there is no landslides the precision of the difference DEM may be high (low noise level) but this noise level may very well change between the landslide zone (with steep topography even if not very tall) and the valley to the North. Additionally the author compare the biggest landslides to the mean noise. . .They should compare to mean landslide depth. Reply: Thank you for the reviewer's comments. There are indeed different precisions according to the DEMs. The studies using scaling laws to derive landslide volume include many large errors, but many papers have been published. We are using higher-resolution data and results, which is clearly shown in the manuscript, but the reviewer seems not to believe it. We have done our best to explain our study and try to show more reliable results to the reader. However, there are also many "mission impossible" tasks in scientific research.

L212 : How were shallow and deep-seated landslide classified ? Clearly I would not call deep seated a slide with a 1-2 m depth, while some of them have 10cm. . . And 10m is not shallow and except in some place most likely much deeper than the soil layer. Reply: Thank you for the reviewer's comments. I am not sure where the reviewer found that deep-seated landslides are 10 cm. Our results show the overall topographic changes. Even if the reviewer's point is true, it is possible that a deep-seated landslide may have some less deformed shallow parts at the edges.

L234-238 : Here the author acknowledge, very late, the problem of canopy (this should be done in the method !). And then dismiss it on the argument that no systematic error are observed and that low elevation difference exist in the "blank area". Well the reason is simple enough : in the carefully selected area of fig 4b, there is only city, agricultural fields and river flood plain. SO very limited high ranging vegetation, and if the images were taken when fields were denuded it would make sense to obtain overall no elevation change. Reply: Thank you for the reviewer's comments. However, the sequence of the figures should match the contents of the manuscript; from the overall structure of the manuscript, we could not first show the results of Figure 4 and then explain the details in the method section. We need to first show Figure 3. If we included this part in the method, maybe other reviewers would comment that the methods and results are mixed. We hope that the reviewer can understand.

Authors figures Fig 1-4 Overall the author show lithological maps everywhere that are almost not discussed. IN contrast a map differentiating agricultural lands, forest , grass land, and shrub or medium height vegetation would be much more useful and a potential place to start to evauate methodological error related to canopy.

Fig 1: could be an inset. Reply: Thank you for the reviewer's comments. This figure shows the overall tectonic setting of the Chuetsu area within a larger geological context to allow readers to better understand the background of mountain building in this region. Meanwhile, the DEM data covering Japan also show the orogenic features and mountains clearly. We actually tried to show this information as an inset figure, but a very small inset figure could not show the geological background clearly.

Fig 3C : The color scale need to be re adjusted to something like +10 / -10 Same for Fig 4, this stretch hides all details and just show the biggest slides. Reply: Thank you for the reviewer's comments. However, we do not agree with the reviewer. This disagreement is because there are deep-seated landslides, very shallow landslides and non-deformed regions. The scale should not be stretched to show false information by adjusting the real elevation change to +10 to -10. This action would mean that we

removed other deformations between +30 to +20 and -30 to -20.

Fig 5: We need to see the uncertainties on the parameters of the V-A relationships and the associated confidence interval on the plot (along the fitted lines). In any case the biggest issue is that the trend of size with depth do not exist for "shallow" landslides. They tend to be randomly distributed around 10m, that is most likely a methodological error, not acknowledged nor discussed by the authors. Reply: Thank you for the reviewer's comments. We do not understand the point of this comment. The figure is in a log plot because the scatter of landslide depths is so large, but how could the figure be interpreted by the reviewer to indicate ∼10 m?

Fig 8B : No idea what the points are or the shade line and how they are drawn... Reply: Thank you for the reviewer's comments. The points show the average denudation depths of the catchments. This figure is just a very simple plot using X (distance to the fault) and Y values (denudation depth of the catchment).

Reviewer 2ïïjŽ Comments: There are grounds for concern over the handling of canopy effects in this paper. I come to this review late in the process, and can see the notes and discussion between the first reviewer and the authors, which are very useful in providing these comments. In essence, the discussion of canopy effects should be more central to the paper and not have been tacked on as afterthoughts and reactions to the first review. So, while the subject is interesting and a lot of work has gone in to the paper, it does not look robust enough for publication in its present form. It would make sense for the authors to have a complete re-think of their analysis, before considering what to do next. Wherever the paper is taken, it is going to meet the same comments at the review stage. Reply: Thanks for the reviewer's comments, we add the following text in the main text to explain the canopy effect in the main text other than just the response to the reviewer comments. "In order to better avoid canopy effect of the forest area, we derive the landslide volumes using the vertical deformation data within each landslide area, meanwhile, the data within the seriously forested region is not used (Fig. 4). The canopy effect is also avoidable because most of the landslides

did not occurred in the forested region, because the seismic landslides mainly occurred on steep slopes which are favourable for landslides. If there are serious canopy effect, then the positive and negative values of the DEM difference should include the canopy, which should be apparently different, hence, could not be comparable. However, our results also show the summing results of the negative and positive value is almost zero. Meanwhile, the largest landslide clearly showed the source and deposit areas, which occurred in the low mountains composed of late Miocene to Pliocene non-marine sediments (Fig. 4). Meanwhile, the derived landslide volumes also showed consistent results that deep-seated landslides are the main contributor to the landslide volume (Fig. 5). The above results indicate that the canopy effects will not seriously affect our results of the landslide volume, hence the distribution pattern of co-seismic denudation pattern."

The deep-seated landslide inventory as shown in the region the reviewer 1 concerned, which should be of serious canopy effect. From this figure, we could clearly see that the landslide region did not mostly occurred on the forested region but the region of bare earth or with grass.

The clearly shown landslide toe and scarp area also indicates our results is correct, otherwise, if there are serious canopy effect, there would not be clear boundary between negative and positive values within one landslides. Hope this figure could explain why our results are not seriously affect by the canopy. Thanks for your time and efforts.

Please also note the supplement to this comment:
https://www.solid-earth-discuss.net/se-2019-3/se-2019-3-AC3-supplement.pdf